# Geopolymer Materials for Extrusion-Based 3D-Printing: A Review

**DOI:** 10.3390/polym15244688

**Published:** 2023-12-12

**Authors:** Laura Ricciotti, Antonio Apicella, Valeria Perrotta, Raffaella Aversa

**Affiliations:** 1Department of Architecture and Industrial Design, University of Campania, Luigi Vanvitelli, 81031 Aversa, Italy; antonio.apicella@unicampania.it (A.A.); valeria.perrotta@unicampania.it (V.P.); raffaella.aversa@unicampania.it (R.A.); 2Advanced Material Laboratory, Department of Architecture and Industrial Design, University of Campania, Luigi Vanvitelli, 81031 Aversa, Italy

**Keywords:** geopolymer, 3D-printing, additive manufacturing, printability, flowability, rheology

## Abstract

This paper examines how extrusion-based 3D-printing technology is evolving, utilising geopolymers (GPs) as sustainable inorganic aluminosilicate materials. Particularly, the current state of 3D-printing geopolymers is critically examined in this study from the perspectives of the production process, printability need, mix design, early-age material features, and sustainability, with an emphasis on the effects of various elements including the examination of the fresh and hardened properties of 3D-printed geopolymers, depending on the matrix composition, reinforcement type, curing process, and printing configuration. The differences and potential of two-part and one-part geopolymers are also analysed. The applications of advanced printable geopolymer materials and products are highlighted, along with some specific examples. The primary issues, outlooks, and paths for future efforts necessary to advance this technology are identified.

## 1. Introduction

An emerging and rapidly growing field of additive manufacturing (AM) technology (according to ISO Standards [1], AM is a process in which focused thermal energy is used to fuse materials by melting as they are being deposited) is 3D-printing (fabrication of objects through the deposition of a material using a print head, nozzle, or another printer technology, according to ISO Standards [1]) with geopolymers (GPs). Increasing the construction industry’s productivity is the main driver behind the development of 3D-printing technology [2]. Moreover, this technology has great potential to decrease construction waste and improve the freedom of shape in built items [2,3]. According to data from the literature, using 3D-printing in construction can help reduce fabrication waste, labour costs, and production time by 30–60%, 50–80%, and 50–70%, respectively [4]. As a result, although 3D concrete printing is still in its early stages of development compared to other industries, it has garnered significant interest worldwide in recent years.

The first additive manufacturing technique used in building, contour crafting, was created in the middle of the 1990s [5] and is based on fused deposition modelling [2,6]. Later, particle bed printing, which selectively deposits the binder liquid into the powder bed to bond the powder particles [2], which was inspired by the technology of stereolithography [2,7], came into view. The most widely used additive manufacturing technique in the construction sector is extrusion-based 3D-printing, in which the object is built by depositing material extruded from the nozzle layer by layer [1,2,8].

The extrusion-based 3D-printing process is schematically represented in Figure 1.

This method involves extruding geopolymer paste via a nozzle, which is then applied layer by layer to make the result. The mechanical components of such printers are propelled by a kinematic system represented by a gantry or a robotic arm. 

The main binder material is Portland cement, which is between 15% and 45% of the total mix proportion in most 3D-printed concrete [9]. In addition, 3D-printed cementitious materials often include more binder than traditional cementitious materials cast in moulds [10]. Since the production of Portland cement accounts for around 8% of the world’s CO_2_ emissions, the growing use of Portland cement could lead to higher material costs and decreased sustainability [11,12]. To create sustainable 3D-printed concrete utilising either extrusion-based [13,14,15] or particle bed printing [16,17] techniques, additional studies have recently concentrated on the viability of using a greener binder, geopolymer. 

Geopolymers, called ‘inorganic polymers’, are amorphous ceramic systems with an aluminosilicate matrix produced by alkalinising natural or waste substances, such as metallurgical, industrial, urban, and agricultural wastes [18]. The term “geopolymer” describes various alkali-activated materials (AAMs) in the current literature. Still, it is most frequently used to describe systems whose binding phase primarily consists of aluminosilicate and is highly coordinated [19]. The aluminosilicate precursor powder reacts with an activating alkaline solution of sodium and/or potassium hydroxides and silicates to form a ceramic amorphous matrix at temperatures below 100 °C [18].

Their potential application is currently being investigated in a variety of scientific and industrial fields, including civil engineering, the automotive and aerospace industries, nonferrous foundries, metallurgy–ceramics, building retrofitting, waste management, art, and decoration. When employed as cement and concrete components, they can decrease production-related energy use, greenhouse gas emissions, and environmental impacts [18,19].

Because of their complexity and extreme variability, it is difficult to create geopolymer mixtures that meet all the strict requirements of additive manufacturing. The development of extrusion-based 3D-printing technology using geopolymers largely depends on acceptable settings and curing durations for extruded layers and their high adhesion. In addition, the mixture needs to be thixotropic. Thixotropy is a time-dependent property that reflects the state changes of pastes under shear and nonshear forces. For the extrusion 3D-printing process, pumping and screw mixing actions are required before the paste enters the extrusion nozzle. At this time, the paste must exhibit good fluidity (low dynamic yield stress) before extrusion, while excellent stacking characteristics (high static yield stress) must be exhibited in the constructed structure after extrusion. This time-dependent phenomenon is affected by the change in the paste state; hence, thixotropy can be used to describe the buildability and 3D structural performance [10,14,20]. It needs to be a viscoelastic liquid-like paste with specific rheological properties [20] for extrusion as well as display solid-like properties to remain stable after deposition, allowing the development of a self-sustaining structure. The quality (depositability, wettability) and uniformity of the aluminosilicate powder used in the 3D-printing process are crucial, and deviations from these requirements result in an anisotropic phenomenon in terms of linear dimensions and mechanical properties, as well as low strength of printable GPs [21]. Instrumentation problems are just as significant as the recipe components. This holds for the dimensional specifications and printing speeds of commercially available 3D-printers. Although 3D-printing with inorganic binders has been a practice for a while [22,23,24], only in the last three years have scientific publications explaining the use of GPs been actively published.

Our statistical analysis shows the growing interest in 3D-printing of GPs in recent years.

The results obtained from the Scopus database (the data were extracted from a statistical analysis of published articles on the topic “geopolymers for 3D-printing” on 12 April 2023) show that academic interest in geopolymer materials has increased by over 95 percent in the last seven years for 3D-printing applications. 

Moreover, data show that more than 58% are articles, 16% are reviews, 14% are conference papers, and 8% are chapters in books. The countries that had the highest publication activity were China, Australia, and the United Kingdom (only the first three countries are mentioned).

The current study aims to comprehensively examine the development of various 3D-printed geopolymer properties, including their printability, flexibility, durability, and economic and environmental advantages.

It is essential to highlight that only studies focused on extrusion technology have been considered, whereas studies [24,25] have not been considered. The reason for this choice is due to the small number of papers published in this area, and most existing products manufactured using particulate printing are unsuitable for structural applications. 

## 2. Geopolymer Materials: Synthesis and Applications

Geopolymers have recently received great attention and research interest due to their potential as green and low-carbon cementitious material [26]. French academic Davidovits was the first to highlight the potential of geopolymers in the 1970s [27]. In the 1980s, he created the first inorganic geopolymer material by reacting natural minerals containing silicon (Si) and aluminum (Al), such as slag, clay, fly ash, pozzolan, and an alkaline activator under mild conditions (below 160 C) [28]. They are a class of aluminosilicate materials produced by an inorganic polycondensation reaction. The geopolymerisation occurs when solid aluminosilicate precursors are combined with alkaline solutions such as potassium hydroxide (KOH), sodium hydroxide (NaOH), sodium silicate (Na_2_SiO_3_), potassium silicate (K_2_SiO_3_), or highly concentrated aqueous alkali hydroxide. Metakaolin and byproducts of various industrial processes, such as biomass ash, steel slag fly ash, bottom ash, red mud, volcanic ash, waste glass, coal gangue, diatomite, bauxite, high-magnesium nickel slag, and so on, can form the solid precursor.

The “two grinding and one burning” procedure typical of Portland cement production, which involves grinding raw materials, calcining clinker, and grinding cement, is avoided by this method, which also considerably benefits energy saving and emission reduction. For instance, just 0.18 kg CO_2_ was released during the production of 1 kg of geopolymer slurry, or around 1/5 of the amount of ordinary Portland cement, and when compared to traditional concrete, the amount of CO_2_ emitted during the production of geopolymer material was decreased by 26% to 45%.

Moreover, geopolymers can be functionalised, formed as organic–inorganic hybrids, or mixed with other materials to form composites to generate novel materials for cutting-edge technological applications [29,30,31,32,33,34,35,36,37].

The organic phase can be added to geopolymer paste in liquid or solid form, such as powder, fibres, or particles [38,39,40,41,42]. Adding a second liquid to a nonmiscible water geopolymer is particularly difficult because of the chemical incompatibility of extremely polar aqueous and apolar organic phases. The organic component, in particular, can be added in various ways and at various stages of the composite’s production: (i) Direct mode. The solid precursors first dissolve in the alkaline aqueous solution to slurry the paste. The organic phase is instantly incorporated into the slurry while being forcefully mechanically agitated before the system hardens. (ii) Pre-emulsification method. Initially, while the activating solution still lacks solid precursors, the organic component emulsifies. To begin the paste-hardening process, the solid precursor is introduced to the stable emulsion of the organic phase in the aqueous activating solution. (iii) Solid impregnation process. The organic phase is impregnated on a solid powder (either the aluminosilicate precursor or a particular adsorbing powder) and added to the alkaline activating solution before being added to the geopolymer slurry.

The geopolymer reaction mechanism model (see Figure 2) consists of the processes of dissolution–depolymerisation and reconstruction–polycondensation when made from low-calcium aluminosilicate raw materials [43,44].

During alkali dissolution, the chemical bonds of aluminosilicate minerals are broken, causing the minerals to break down into aluminum–oxygen [AlO_4_]^5−^ and silicon–oxygen [SiO_4_]^4−^ monomers. As the dissolution progresses, monomers connect to form dimers, and dimers with other monomers form trimmers, multimers, and so on. Then, a reorganisation and polycondensation occur to form a three-dimensional network structure amorphous sodium aluminosilicate hydrate (N-A-S-H) gel.

Since Al^3+^ dissolves more quickly than Si in the aluminosilicate matrix in the alkaline environment, the concentration of Al^3+^ is higher during the early stages of the reaction (when the reaction achieves saturation). As a result, a metastable N-A-S-H gel with a high aluminium content (Si/Al ratio: 1.0 to 1.3, Al-rich gel I) would precipitate as a byproduct [46]. More Si-O groups dissolve during the reactionmore Si^4+^ concentration increases in the solution and its percentage in the N-A-S-H gel (Si/Al ratio about 2, Si-rich gel II). The silica–alumina skeleton and a metal cation with a balanced charge determine the polymer’s three-dimensional network structure, and this structural rearrangement defines the material’s pore shape and distribution. High temperatures increase the system’s kinetic energy, which improves intermolecular attraction and the efficiency of solute molecular bond breaking by solvent molecules [47]. As the medium in which the reaction happens is overlooked when the activator level is below a particular threshold, numerous aluminosilicate precursors remain unreacted [48]. In the medium where the reaction occurs, numerous aluminosilicate precursors remain unreacted if the activator concentration is below a particular level. A reduced dissolution of aluminosilicates does not result in a changed setting time.

There are two types of geopolymer application fields: those with standard physical and mechanical properties, as well as those with functional and advanced properties.

In the first category, geopolymers are used in building, construction, repair, restoration, maritime construction, pavement foundation materials, 3D-printing, high-temperature and fire-resistant materials, and thermal and acoustic insulation. Some specific applications include heavy-metal pollution immobilisation, pH regulator materials, catalysts, conductive materials for moisture sensor applications, and thermal storage [13,49,50,51,52,53,54].

Buildings in specific industries, such as firefighting structures, insulation walls, and nuclear power plants, might benefit from functional applications. These include fire control, isolation, heat preservation, and hazardous ion adsorption [55,56,57,58,59,60,61,62].

The following are the main advantages of geopolymers:-High strength and rapid hardening. Due to a rapid polymerisation rate, the three-dimensional network structure of geopolymer materials is simple to produce. For instance, after curing for 4 h at room temperature, metakaolin-based geopolymers’ compressive strength can reach 15–30 MPa and continues to increase over time [19,63,64];-High-temperature resistance and good thermal stability. A compact oxide network structure protects the inner material from oxidation in the air. With a linear shrinkage of 0.2% to 2% and preservation of more than 60% of its initial strength at 800 °C, geopolymer exhibits higher temperature mechanical strength and more thermostable behaviour than Portland cement [65,66,67,68,69];-Strong capacity to bind at material interfaces. Ordinary Portland cement has a poor interfacial binding force because it is susceptible to calcium hydroxide enrichment. The transition zone is preferred and orientated at the interface where it is coupled with aggregate. The final product of the geopolymer materials is primarily a three-dimensional network gel with covalent bonds tightly bound to the aggregate interface and strong bonding ability [70,71]. There is no hydration reaction of calcium silicate in the geopolymer materials;-Good durability and corrosion resistance. The geopolymer has superior resistance to sulphate corrosion since the hardening process does not produce sulphate minerals like ettringite. The geopolymer breakdown rate in a 5% solution of sulfuric acid or hydrochloric acid is just around 1/13 or 1/12 of the rate of Portland cement under the same conditions, demonstrating good stability in acid conditions. Geopolymers can create compact structures that have high durability and impermeability [28,72,73,74,75,76,77,78].

All these exciting features encourage geopolymer use in additive manufacturing, for environmental, technological, and commercial reasons (see Figure 3).

However, there are still unsolved issues and challenges with geopolymer materials, as listed below: (1) Although there are a variety of raw materials used to obtain geopolymers, including solid wastes like metakaolin, fly ash, volcanic ash, coal gangue, etc., the instability and volatility of solid wastes as well as the differences between raw materials have a significant impact on geopolymer properties; (2) Treatment during the processing and deposition of geopolymer pastes results in high temperature dependency of the polycondensation reactions and the correlated rapid hardening and increase of the slurry viscosity, which results in poor workability and unreliable setting times [20]; (3) Brittleness and toughening modification of geopolymer materials are presently under investigation. (4) Geopolymer activators are primarily alkaline activators such as Na_2_SiO_3_ and NaOH, which are limited and costly, contributing to the high cost of geopolymer material. (5) The geopolymer product and technology evaluation system has yet to be established, and it lacks satisfactory uniform standards and specifications. These factors severely restrict the use of geopolymers on a large scale.

## 3. Characteristics for Geopolymer Printability

Geopolymer materials should possess many characteristics to meet the critical parameters for 3D-printing technology, such as rheological, physical, and mechanical properties (see Figure 4). The key parameters include aluminosilicate raw material and activator compositions, reinforcement fillers, printing variables, curing conditions (temperature and time), and post-processing of the 3D-printed specimens.

One of the most critical parameters is the composition of the raw materials. To meet the printability characteristics in addition to the aluminosilicate source material and alkali silicate/hydroxide activating agent, it is possible to include plasticisers, accelerators or retarders of hardening, thixotropic thickener, and other components, as shown in Figure 5.

Aggregates such as quartz, accelerators or setting retarders, plasticisers, and so on can be added as additives. Their characteristics will be discussed in detail in the following sections.

Reinforcing components (fibrous or dispersed) are also employed in the mixture to minimise shrinkage deformations and increase the hardened material’s fracture toughness and mechanical properties (see Figure 5).

The raw materials for the printed geopolymer are chosen depending on the product operating conditions, target functional features, and the features of the 3D-printing method.

### 3.1. Aluminosilicate Raw Materials

The most utilised aluminosilicate source for 3D-printed GPs is low calcium (Class F) fly ash from coal combustion. Table 1 reports the mixture compositions of the extrusion 3D-printed geopolymers.

This is due to Class F fly ash’s widespread availability and high SiO_2_ and Al_2_O_3_ oxide content [97,98,99]. Ashes with a high CaO concentration (Class C) are less appropriate for these applications due to their excessively quick setting, which interferes with the production of geopolymer gel (sodium aluminosilicate hydrate, N-A-S-H) [100]. With the application of high concentrations of NaOH, the setting time and degree of geopolymerisation can be increased [101]. Fly ash can be used alone as a precursor to 3D-printing GPs [80,86,91,94] or in combination with other industrial wastes and byproducts [13,21,28,53,81,82,83,88,89,90,92]. 

The poor reactivity of Class F fly ash can be overcome by adding blast-furnace slag (BFS) with amorphous character and high solubility in an alkaline medium [101]. Partly substituting fly ash with blast-furnace slag (up to 15–40 wt%) in 3D-printed GP mixes increases yield stress, thixotropy, and compressive strength [79]. The BFS contribution does not considerably alter the change in the rheological characteristics of geopolymer pastes at low replacement levels (up to 10 wt%). However, the early age compressive strength modifications more dramatically affect the homogeneity of the GP microstructure [14]. In the printable geopolymer mix, silica fume (SF), also known as micro silica, is used as an additional source of amorphous SiO_2_ [13,53,88,89,90,92] (see Table 1). SF, along with BFS, is primarily used to replace FA in partial replacement and helps to manage the rheological parameters of GPs. The presence of reactive silanol groups on the surface of the SF particles, as well as their fine particle size (95% of particles smaller than 1 μm [102], increase the yield stress and viscosity of printed geopolymers in the fresh state [14]. SF is also used to improve mixture cohesion [14].

Thanks to the stable composition and low impurity level of the dehydroxylated form of the clay mineral kaolinite (metakaolin, MK), its use [17,85,95] as a precursor for 3D-printable geopolymers improves the reproducibility and interpretation of the experimental results and their modeling [20]. Metakaolin has an Al_2_O_3_ and SiO_2_ concentration of 90–95% [103]. In an alkaline environment, MK has a high dissolving ability (metakaolin > slag > fly ash > kaolin [104]), and the resultant GPs have a homogenous aluminosilicate phase [103].

### 3.2. Alkali Activators

The fundamental component of geopolymers in the alkaline activating reaction is alkaline activators [105]. These considerably affect the rheological properties of geopolymers. In addition, the presence of alkaline activators is critical in the kinetics of the geopolymerisation.

An alkaline activator typically comprises alkali metal hydroxide and alkali metal silicates to activate geopolymers. NaOH and Na_2_SiO_3_ are two of the most commonly utilised because of their inexpensive cost and outstanding rheological performance [106]. Potassium-based geopolymers have a higher yield stress than sodium-based geopolymer systems [106]. The Na_2_SiO_3_ activating systems, on the other hand, are consistent with the deflocculating action in particle suspensions [107]. As a result, the amount of hydroxide and silicate in the geopolymer mixture is carefully controlled to get the best Si/Na or SiO_2_/Na_2_O ratio. Because sodium ions firmly connect and retain the hydration layer of water molecules, an increase in Na ions in the solution promotes the dissolution of aluminosilicate minerals in the mixture. They are less likely to interfere with condensation and reduce free water [108]. As demonstrated in the literature [79], increasing the alkali activator increases the geopolymers’ yield stress and apparent viscosity. A higher alkali activator amount results in increased pH, higher yield stress, and ionic strength of surface charge [109]. In addition, an enhancement of the activator’s molar ratio (MR) leads to an increase in the apparent viscosity and yield stress of the geopolymer mixture, which is mainly due to an increase in the activator’s viscosity [110].

### 3.3. Aggregates and Fillers

3D-printed geopolymer mixtures, like cementitious materials, can incorporate aggregates, the most common of which are silica (quartz) sand (see Table 1). Additive manufacturing technologies limit the size of fine aggregates. They primarily employ fine sand with a median particle size (d50) of 170–250 µm [17,21,86,111,112]. This is critical for obtaining improved powder bed homogeneity in powder-based 3D-printing and maintaining a narrow extruder nozzle diameter in extrusion-based 3D-printing. Sand incorporation in printed geopolymer mixes decreases the slump rate [111]. The constant gradation of various types of sand might increase mixture extrudability [113]. Bong et al. [113] used the gradation approach based on the Fuller–Thompson theory, mixing finest, medium, and coarsest sands with D50 values of 176, 498, and 840 µm, respectively. When used as an aluminosilicate source of fly ash and slag, 3D-printed GPs holding sands with this particle size distribution improved the buildability of the mixes. Weng et al. [114] observed a similar result with OPC-based 3D-printing materials. Chougan et al. [93] and Muthukrishnan et al. [112] have employed varied fractions of sand in the composition of printed geopolymer mixes. Calcium carbonate (CaCO_3_) and other alternative aggregates are also suitable (see Table 1). Their use can help to decrease shrinkage and cracking in geopolymer systems [84].

The rheology of fresh geopolymer mixes is also influenced by reinforcement fibres or particles. Variation in fibre content in the combination has a significant impact on the SYS and PV, as shown in Table 1. Various reinforcement types (RTs) such as polypropylene (PP), polyvinyl alcohol (PVA) [114], polyethylene (PE) [115], and nano-graphite particulates (NGPs) [116], attapulgite [117], benzobisoxazole (PBO), and microcrystalline cellulose (MCC) have been commonly used to improve the fresh and hardened characteristics of geopolymers for 3D-printing. The overlapping and mechanical interlocking of the needle-like fibres with the matrix material improves the SYS when a sufficient number of wollastonite fibres are used. Furthermore, the breakdown of wollastonite fibres in an alkaline solution raises the concentration of Si and Ca ions in the system [118]. This increase in Ca ions in the solution system accelerates the geopolymerisation process, increasing the combination’s SYS [119]. The number of reinforcing fibres must also be adjusted to improve yield stress and plastic viscosity [9]. It has been demonstrated that a higher proportion of reinforcing fibres increases yield stress and viscosity, requiring torque above the acceptable range [120]. Shear stress increases from 0% to 1% with reinforcement fibres; additional reinforcement increases result in printing failure [120]. However, different reinforcement fibres provide a different percentage of reinforcing to achieve superior rheological properties [121].

### 3.4. Additives for Printability

The use of active organic and mineral additives in the formulation allows the rheological characteristics of printed geopolymer mixes to be controlled. Plasticising additives have a unique position in them. Surface active functional groups and finely distributed fillers with a high specific surface area considerably influence the geopolymer mortar’s viscosity. Sun et al. used ball milling to produce a surface modification of hexagonal boron nitride, producing many hydroxyl groups on its surface. When reacting with alumino–silicate oxides, such fillers can both decrease (at low levels) and enhance (at high contents) viscosity and viscoelasticity in geopolymer mortar [116]. Urea, naphthalene, and polycarboxylate exhibit the qualities of a superplasticiser for 3D-printing geopolymers [122]. Because of its flocculation properties, attapulgite clay can be employed as a thixotropic thickener to increase the yield stress of geopolymer blends [81,86]. Magnesium alumino-silicate also improves the thixotropy of new geopolymer pastes [123]. CMS (sodium carboxymethyl starch) is a viscosity modifier and retardant [84]. Increasing CMS content increases water retention and the setting time of geopolymer paste.

CMS (sodium carboxymethyl starch) is a viscosity modifier and retardant [84]. Increasing CMS content increases water retention and the setting time of geopolymer paste. Furthermore, using CMS in a 3D-printed geopolymer composition might minimise hardened product drying shrinkage by decreasing the aluminosilicate precursor’s polycondensation and reducing the polymerised gel product quantity [84]. Sucrose in solid form can be used as a retarder to extend the mixture’s setting time [113,123]. Sucrose’s influence on the dynamic viscosity of geopolymer paste is negligible due to its great solubility in water [123]. Graphene oxide (GO) may significantly alter the rheological characteristics of geopolymeric ink mixtures [84,85], implying a substantial interaction between GO and GPs. The hydrophilic properties of graphene oxide and aluminosilicate particles, according to Zhong et al. [124], explain this phenomenon. Adsorbed water molecules on their surfaces produce a closed lamination layer between GO and aluminosilicate particles. As a result, such a system can endure elastic deformation, contributing significantly to the complex modulus. Nano graphite [93] and polyethylene glycol [95] are also used as rheological additions in geopolymer blends to improve workability, flowability, and form stability.

## 4. Fresh-State Properties

### 4.1. Rheology

Flowability is a crucial characteristic of a fresh geopolymeric paste in extrusion-based 3D-printing, defining its ability to flow under pressure in the printing system. Flowability relates to rheological viscoelastic parameters, particularly yield stress (associated with the elastic and storage response of the paste) and its dynamic viscosity (related to the dissipative viscous character of the fresh geopolymer). High values of these viscous and elastic components decrease flowability while avoiding the collapse of the 3D-printed part [86,108,114]. Organic and inorganic modifiers can be utilised to modify viscosity. However, often, a negative impact on buildability has been observed [20]. As a result, determining rheological characteristics necessitates striking a compromise between flowability and buildability to provide a high-quality geopolymer 3D-printing material. Internal parameters such as source materials, type, and activator content all have a substantial impact on the viscosity of geopolymer mixtures [113]. Ma et al. [113] measured slump rates of Na-, K-, and Cs-based geopolymer inks and found that those with a sodium-based activator had higher geopolymer viscosity than those with potassium or cesium-based activators. This behaviour is dictated by Na^+^’s smaller ionic radius, which binds molecules of H_2_O more strongly. Sahin et al. [115] discovered that increasing the molarity of NaOH solution from 5 to 11.25 M increased the flowability index of geopolymer mixes based on construction and demolition debris, which was associated with an increase in precursor dissolving velocity. The geopolymer combination became sticky as the molarity of NaOH increased, which was explained by the rising viscosity of NaOH solutions and the limits of ion mobility in the mixtures. The flowability of fresh geopolymer mixes is also affected by the types and amounts of aluminosilicate precursors and aggregates. This mainly refers to changes in interacting with alkali sources, as well as particle structure and size [83,119]. The use of reinforcing fibres increases the viscosity of 3D-printable geopolymer composites [120]. Extrudability describes the ability of a geopolymer combination to travel smoothly into the nozzle of a printing head, generating a continuous filament. The combination should have a low plastic viscosity and optimal yield stress for smooth extrusion. A poorly balanced or insufficient mixing might result in particle segregation, nozzle lodging, or filament tearing [82,122,125]. The Si/Na ratio of the alkali activator has a substantial impact on the extrudability of 3D-printable geopolymers, with increasing viscosity and yield stress [126]. According to Zhang et al. [126], decreasing this ratio is desirable in terms of enhancing the ability of structure rebuilding (specifically, changing particle linkage). Extrudability can be improved by adding urea, CMC, and other rheologic additives [82,125]. Panda and Tan [89] discovered that high static yield stress reduced the extrudability of geopolymer mixtures with a high sand/binder ratio. To commence the flow of such mixtures, substantial pumping pressure was necessary. Fine aggregate particle size distribution is also critical. The design of 3D-printing materials based on Fuller–Thompson’s theory with a steady gradation of several types of sands demonstrates good extrudability of fresh geopolymer mixes [113]. Extrusion velocity, size, and shape of the printing nozzle generating its geopolymer layers are essential printing process characteristics [82,122].

### 4.2. Buildability

Another critical parameter of the extrusion 3D-printing process of a new geopolymer mixture is buildability, as this determines the shape and buildability stability of 3D-printed systems [127,128]. As fibre content increases, so does the ability to maintain form [91]. However, the adhesive strength between the layers deteriorates as well [91,129]. The aluminosilicate supply, reinforcement elements, and fillers all affect the slump rate, which is used to assess the form stability of the printed layers. Archez et al. [111] showed that nonreactive fillers, particularly argillite, reduce the liquid-to-solid ratio contribution. The addition of reinforcement materials (such as glass fibres and wollastonite) and metakaolin to the geopolymer mixture can also reduce the slump rate value, resulting in a change in the viscosity of the geopolymer paste and an acceleration of the geopolymerisation reactions [111]. The slump rate is affected by printing settings as well as mixture composition. A decrease in printing speed contributes to a decrease in droop because it implies a longer time between layers [111]. The time interval, however, should be adjustable because the start of polycondensation processes can limit interaction with the following layer.

### 4.3. Open Time

Open time describes the period during which a fresh geopolymer mixture has a suitable consistency for printing. On the one hand, the optimal available time must be short for the geopolymer material to quickly gain sufficient strength to withstand successive extruded layers without significant deformation. On the other hand, it must be long enough to ensure that the neighbouring printed layers are linked together and do not develop cold joints [93]. The open time of printed geopolymers is less than the first setting time of traditionally mould-cast GPs [83,89,113]. The type and dosage of the aluminosilicate precursor and alkaline activator used could have a large impact on the open time and setting time [130,131,132,133,134,135,136]. When the concentration of reactive calcium sources, such as BFS, is increased, the open time decreases, resulting in the formation of calcium silicate hydrate (C-S-H) and calcium aluminosilicate hydrate (C-A-S-H) gels [83,89,93]. Heating increases the open time of printed geopolymers [128]. Several retarders and accelerators [113,123,137] can be utilised to control the open time. After reviewing several papers, it was discovered that various authors recommended optimum values for the spread diameter, slump, open time, and setting time. These studies define a specific range for the open time and set times based on the optimum values identified in their tests. Available and initial setting times of 20–80 min and 22.9–90 min, respectively, are reasonable standards for future research for producing small components manufactured with extrusion-based 3D-printing of geopolymers. A good range for the slump has been observed to be 8.8–20 mm for assessing the buildability, based on the recommended values from the literature. In the various papers, the workability and printability of geopolymers are also analysed for 3D-printing based on spread diameter or SFD. An adequate SFD range of 120 to 206 mm was obtained based on their recommended values. The analysis revealed that the recommended response values of the various studies are not universal; they may vary depending on the composition of the concrete materials. For example, higher FA values are preferred in some cases, whereas lower FA values are preferred in others. Dedenis et al. [137], reported that a combination of FA and metakaolin FA and MTK increased the fresh and rheological properties of the paste, and the optimal mix (10 to 15% of metakaolin, 30% of fly ash) proved to be a combination of both components in terms of workability, extrudability, yield stress, cohesion, stability, and printability. It also reduced bleeding and segregation and improved stability. In addition to FA, this is determined by the typology and amount of other constituents used in the fresh mixture. On the other hand, each mineral’s overall effects on the rheological, physical, and mechanical properties discussed in the literature are undeniable. For example, due to the spherical shape of the particle, which acts as a ball bearing and increases the fluidity of the mixture, adding FA reduces the yield stress of the fresh mixture. Unless it chemically reacts with some minerals in the mixture and produces some distinct compounds, including FA always serves the same purpose as discussed above. Furthermore, it has been determined that reinforcement fibres and alkali activators significantly impact the rheological and physical properties of geopolymers. However, there needs to be more research on the rheological requirements of geopolymers for 3D-printing.

## 5. Hardened-State Properties

### 5.1. Density

When the mixture design or printing process settings are ineffective, 3D-printing with extrusion contributes to lower geopolymer density than casting due to under-filling samples [83,89,93]. Defects or heterogeneities in the extrusion-based printing process create voids in the cured GPs. The existence of gaps in 3D-printed geopolymers can reduce their mechanical strength [93]. Extrusion at high-pressure aids in improving the density of printed geopolymers [83,89]. Alternatively, the density of the printed geopolymer material in powder-based 3D-printing is determined by the degree of pore saturation of the aluminosilicate powder precursor layer with an alkaline solution [17]. In this scenario, the GP density increases with the degree of saturation of the powder bed until a certain point, after which the dimensional accuracy degrades due to the amount of liquid in the sample, as demonstrated by Voney et al. [17].

### 5.2. Mechanical Properties

Mineral types, reinforcement, alkaline activators, printing parameters, and post-processing of printed specimens all have a direct impact on the mechanical properties of geopolymers. The effects of these parameters on compressive strength (CS), flexural strength (FS), interlayer bond strength (IBS), tensile strength (TS), and Young’s modulus (YM) have been investigated individually and the actual phenomena behind these effects have been investigated. Controlling the mechanical properties of printed geopolymers is more difficult than controlling OPC due to the numerous internal and external parameters involved, as shown in Figure 6.

Because of printing process peculiarities, the mechanical strength of 3D-printed geopolymers exhibits anisotropic behaviour depending on testing direction [13,22,25,79,82,88]. The laminated method of manufacturing in an extrusion-based process yields the following results: The longitudinal direction (specifically, the direction of printing) always has greater strength than the lateral or perpendicular directions [25,82]. The mechanical anisotropy of the geopolymers created by powder-based 3D-printing is caused by the preferential orientation of the powder particles and the weak interlayer adhesion [22,138].

When compared to casted geopolymers, the mechanical properties of 3D-printed GPs are degraded by the presence of voids and weak interlayer bonding [17,83]. However, several studies demonstrate that comparable strength values can be achieved by controlling printing parameters, along with a thorough material design and dying mode optimisation [83,93,139]. Heat treatment increases the mechanical strength of printed geopolymers compared to room temperature-cured samples [80,94,139], as seen in casted geopolymers [140]. 

According to Bong et al. [82], Na-based printable GPs that are used as an aluminosilicate precursor granulated ground blast-furnace slag and fly ash have a higher compressive strength than K-based GPs, regardless of the type of silicate solution or the solution mass ratio to hydroxide solution. Taking into account the findings of related research on cast geopolymers, this conformance may change when different precursors are utilised [141]. GGBS increases the compressive strength of 3D-printed geopolymers at an early age, resulting in a more homogeneous microstructure [14]. According to Bong et al. [82], independent of silicate solution type or solution mass ratio to hydroxide solution, Na-based printable GPs utilised as an aluminosilicate precursor granulated ground blast-furnace slag and fly ash have greater compressive strength than K-based GPs. 

The significant improvement in the strength of 3D-printed geopolymers allows for continuous cable reinforcing [92,142]. According to the mechanical test results of Li et al. [142], embedding a continuous metal micro-cable (diameter 1.2 mm) in printed arching beams and reticular spiderweb-like geopolymer structures increases their bending capability by 260 percent and 132 percent, respectively. As reported in Figure 7a,b [142], the load–displacement curves of the reinforced arched beam and the corresponding nonreinforced beam that were subjected to a similar bending load exhibited different failure patterns. Similar effects have been seen in previous investigations [90,92,106]. Incorporating short fibres also allows for increased flexural strength [129,143]. As shown in Figure 7c,d [129], the introduction of fibres in the layering process can impact the bond strength between subsequent layers. The distribution and orientation of the fibres are essential in this scenario. 

Flexible fibres (natural and polymer) are significantly superior to steel, which might cause fibre protrusion between layers and weak interlayer bond strength [129]. Sun et al. [84] have shown that changing admixtures used to regulate the rheological characteristics of a fresh geopolymer slush can negatively affect the mechanical properties of printed geopolymers. They used sodium carboxymethyl starch (CMS) as a modifier of a slag-based geopolymer for 3D-printing; it aided in the reduction of compressive and flexural strength as its concentration increased. Because of the high viscosity of the geopolymer slush and the development of a screening dense polymer film on the surface of silicate powder, it was connected to the aeration effect, CMS resistance to dissolve, and the diffusion of raw materials [144]. Ma et al. [119] used extrusion 3D-printing technology developed by G. Lazorenko and A. Kasprzhitskii to create lightweight bio-inspired Bouligand structures with high strength, toughness, and nonbrittle failure behaviour. The best results were obtained when three wt% short carbon fibre was used. The flexural and compressive strengths of such geopolymer composites were 309.2 percent and 375.8 percent greater, respectively, than those of nonreinforced GPs. Aside from short carbon fibre, SiC whiskers improve the strength and toughness of geopolymer composites with complicated hierarchical designs [145].

### 5.3. Shrinkage

Drying shrinkage is intimately related to the long-term dimensional stability and crack formation of 3D-printed geopolymers. Although several studies have been conducted on shrinkage in traditionally mould-cast geopolymers [146,147,148,149], comparatively few studies have been conducted on shrinkage in printable GPs. Because of the lack of formwork, the printed geopolymers’ drying shrinkage has a vast surface area. This promotes significant water evaporation rates, which is apparent in the shrinkage of 3D-printed GPs throughout the drying process. It is widely understood that modifying agents, such as sodium carboxymethyl starch (CMS) [84], deter shrinkage value.

Short fibres are also helpful in reducing shrinking [111,150]. J. Archez et al. [111] studied the influence of high temperatures on the shrinking of 3D-printed geopolymer lattices. The volumetric shrinkage of the metakaolin-based GPs rose by 20% after thermal treatment at 300 °C compared to the reference sample cured at 80 °C for 1 day, which was ascribed to the total loss of bound water. Short fibres are also helpful in reducing shrinking [111,150]. Printed geopolymer matrices incorporating 20% siliceous sand as filler exhibited less shrinkage but more pronounced breaking. The researchers attributed this behaviour to the expansion of sand filler when heated, which prevented shrinkage of the geopolymer matrix and created stresses, resulting in crack formation. The printed geopolymer matrix shrank the most after being thermally treated at 700 °C. In this scenario, the volumetric shrinkage of the printed geopolymer structures without filler and samples with sand increased by 30% and 25%, respectively, as compared to reference samples.

### 5.4. Layer Adhesion

In extrusion-based 3D geopolymer printing, providing adhesive strength between neighbouring layers is a critical criterion for final product stability. Interfacial bond strength is directly affected by printing speed and the time gap between applying additional layers [53,111]. Short time intervals allow for better material mixing, resulting in a strong links between the layers. On the other hand, long intervals cause an interface to form and limit the interaction of adjacent layers due to the onset of the polycondensation process. The adhesion between the layers is harmed by the reinforced fresh geopolymer mixture’s increased stiffness, which includes fibres [91,129,151]. It is preferable to align the threads parallel to the direction of printing [91,111]. The ambient parameters (temperature and relative humidity) are also important. For 3D-printed GPs based on metakaolin, Archez et al. [111] showed that decreasing the temperature from 20 °C to 8 °C while concurrently raising the RH from 43 to 83 percent deteriorates the interface between the layers and restricts the possibility of material consolidation over time, leading to a material integrity violation.

### 5.5. Durability

There needs to be more research on the durability of 3D-printed geopolymers. As with OPC-based printed materials [152], the penetration of layer interfaces is a vital issue influencing the durability of 3D-manufactured GPs. High porosity in the interlayer zone of printed geopolymer structures makes them less resistant to freeze–thaw attack, aggressive sulphate and chloride ion penetration, carbonation, and other effects. As a result, these features necessitate in-depth research to better understand the transport processes and long-term durability of 3D-printed geopolymers.

## 6. One Part Geopolymers

Usually, geopolymer is a two-part mixture made by combining an alkaline solution with precursors rich in alumina and aluminosilicates, such as fly ash and metakaolin. Laterite, a promising mineral with a high abundance and good performance, was recently used as the precursor [153]. Alkaline activators such as alkali hydroxide, silicate, and aluminate activate the precursors [154,155,156,157]. High compressive strength, fire resistance, rapid hardening, salt and acid resistance, and other environmental benefits distinguish geopolymer from ordinary concrete [158,159,160]. Despite the advantages of a two-part geopolymer, there are several issues with viscosity and managing dangerous alkaline activator solutions for large-scale printing. As a result, using a solid activator to create a one-part geopolymer can help address some of these issues [12,161,162]. In addition to water, a dry mixture of a solid alkali-activator powder and a solid aluminosilicate precursor is required to make one-part mixes.

In a one-part geopolymer, a solid activator can be any material that raises the pH of the reaction mixture, offers alkali cations, and aids in dissolving [163]. Anhydrous sodium metasilicate and grade sodium silicate are the most commonly utilised solid activators [164]. Solid activators have various advantages over liquid activators, including being easier to handle on-site, being free of toxic highly alkaline liquids, and being produced at a lower cost with a lesser environmental impact. Incorporating a solid activator facilitates mixing processes similar to OPC, in which the solid ingredients are dry-mixed before adding water (see Figure 8). Due to the tiny diameter of the various nozzles utilised in extrusion-based 3D-printing, fine aggregate with particle sizes less than 2 mm is primarily employed for geopolymer 3D-printing [165]. Fine aggregate incorporation is restricted, with an aggregate-to-binder ratio of 1.2–1.9 for geopolymer 3D-printing [89]. The addition of additives to the mixture allows for the adjustment of the rheological properties of the printed mix, changing its fresh and hardened properties. 

### 6.1. Rheology

One part-geopolymer flowability and extrudability are also dependent on the rheological viscoelastic behaviour of the paste base materials and their formulation; an ideal viscoelastic mix should have a sufficiently viscous character to allow good flowability combined with a level of elasticity that could guarantee a suitable yield stress for extrusion [166]. Muthukrishnan et al. [123] measured the static yield strength with time to explore the effect of increasing the activator content on the flowability of GGBS and FA one-part geopolymers. It was discovered that increasing the activator content resulted in a faster evolution of yield stress, requiring more pumping energy. Several papers have investigated the influence of various precursor materials and discovered that loading the mix with calcium-containing elements, such as GGBS, reduces extrudability. Guo et al. [86] found that including up to 10% SF in FA-based one-part geopolymers improves particle packing and increases viscosity, while increasing the SF ratio decreases viscosity due to its tiny particle size. It was discovered that increasing the activator content resulted in a faster evolution of yield stress, requiring more pumping energy. As a result, employing more than 10% SF in the mixture may result in decreased extrudability due to significant viscosity loss. Shah et al. [167] discovered that increasing the amount of GGBS in the combination decreased flowability, which could be owing to the existence of more nucleation sites at the early stage that increased calcium in GGBS provides, resulting in the mixtures hardening quickly [167]. Furthermore, the angular shape of the GGBS tends to reduce workability [168]. The effect of substituting the precursor and aggregate individually with wollastonite powder on the workability of FA- and GGBS-based one-part geopolymers was investigated by Bong et al. [169]. The results demonstrated that increasing the replacement level reduces the spread diameter by producing network structures that can resist flow due to wollastonite’s needle-like morphology. In another study, Bong et al. [170] printed five layers of square slabs with a total length of 4810 mm for each layer to explore the effect of replacing fine natural sand with wollastonite microfibre on the extrudability of FA- and GGBS-based one-part geopolymers. The mixture with a 10% replacement level was found to have equivalent workability to the reference mix, and both the reference mix and the mixture with a 10% replacement level were effectively extruded. Flowability, on the other hand, can be enhanced by raising the water/binder ratio. However, exceeding a particular water content threshold may result in segregation and pipe blockage [171,172]. Incorporating retarders, on the other hand, can improve extrudability by slowing reactions [167]. In this study, it was reported that a retarder amount of 4% improves the early age strength but also reduces the workability and setting time of one-part geopolymer mortar. Thus, this is recommended as the optimum way to obtain early strength requirements. Cheng et al. [173] studied the influence of several types of superplasticisers (specifically, polycarboxylate (PC), melamine (M), and naphthalene (N)) on the flowability of calcium carbide residue-waste red brick powder-based one-part alkali-activated materials. The researchers discovered that adding 1.5 percent of PC to the various types considerably boosted the flowability of the mixtures to a level comparable to the OPC mixture. Similarly, Alrefaei et al. [174] investigated the influence of PC, M, and N superplasticisers on the mini-slump performance of FA-GGBS-based one-part geopolymers. They discovered that the flowability of the mixtures increased, with polycarboxylate showing the greatest improvement.

Regarding the rheological characteristics, Panda et al. [79] found that because of the calcium-rich chemical composition of GGBS, increasing the GGBS concentration in an FA-based one-part geopolymer improved yield stress, plastic viscosity, and viscosity recovery. The thixotropic quality of the mix is enhanced due to the increased packing density caused by the angular morphology of the GGBS particles. The rheological behaviour of geopolymers based on lateritic clay (LAC) and iron-rich laterite clay was examined by Kaze et al. [175]. LAI as iron (Fe) has a higher reaction rate than Si and Al, the results indicated that LAI had a much higher yield stress than LAC. This is explained by the more significant deformation that LAI displayed as a result of the higher interaction rate between its constituents. Iron hydroxide gel is thus produced by the rapid precipitation of iron species in an alkaline medium, which accelerates the polycondensation process and gives the gel a stiffer structure. The impact of varying calcination temperatures on the rheology of a geopolymer based on meta-halloysite was examined by Kaze et al. [176]. They found that by raising the calcination temperature, more reactive phases were formed in the geopolymer, which improved its rheological behaviour. Ma et al. [177] showed that adding steel slag in place of up to 100% of the FA in a one-part geopolymer based on GGBS and FA decreased the combination’s rheological properties. The decrease may have been caused by the steel slag’s low reactivity, which prevented GGBS and FA from reacting with the activator to form the hydration gels. Moreover, Bong et al. [170] reported that adding wollastonite microfibres in place of 10% of fine sand in a one-part geopolymer based on FA and GGBS increased yield stress and reduced plastic viscosity. Physical interlock and overlap between wollastonite acicular particles could explain the higher yield stress. At the same time, the decrease in plastic viscosity may be attributable to wollastonite’s longer particle form relative to the mixture’s other solid particles. Furthermore, it was discovered that replacing 10% of the sand with fibres somewhat improved its thixotropic property and recovered 80% of the viscosity. Aside from the effect of different precursor materials, activator content influences the rheological behaviour of the mix. Muthukrishnan et al. [123] discovered that increasing the activator level increased the mixture’s yield stress, viscosity, and thixotropy [123]. Higher activator percentages, on the other hand, may reduce the plastic viscosity of the solution [79]. The composition of the activator (i.e., alkali modulus) also has a significant impact on the rheological properties of the geopolymer. According to [178,179], raising the SiO_2_/Na_2_O ratio increased the geopolymer mixture’s viscosity and yield stress.

### 6.2. Mechanical Properties

Several investigations have demonstrated that equivalent strength values can still be produced by modifying the printing conditions and mix design for hardened one-part geopolymers [93,139]. The strength of a one-part geopolymer is generally more robust when a calcium-rich precursor is used, depending on the mix design. The more the GGBS incorporation, the greater the compressive strength [93,180]. However, raising GGBS above a certain limit reduces compressive strength, possibly attributable to a lack of workability. Panda et al. [79] discovered that raising the GGBS content from 15% to 40% enhanced the compressive strength values of an FA-based one-part geopolymer due to the early synthesis of C-S-H. They also discovered that increasing the activator dosage from 10% to 20% increases strength values because more Si ions are accessible for geopolymerisation. Choosing the best precursor proportion for a given activator percentage and water volume can improve compressive strength [181]. Dong et al. [181] discovered that the materials’ fineness substantially impacts the geopolymer’s mechanical strength, with a finer activator greatly increasing the compressive values of one-part geopolymers. Moreover, compressive strength is influenced by the kind of activator used; sodium silicate is the most popular and efficient solid activator. According to Ma et al. [182], partially substituting Na_2_SiO_3_ for Na_2_CO_3_ decreased the compressive strength of one-part geopolymers because the presence of Na_2_CO_3_ decreased the degree of geopolymerisation. The value of compressive strength can be lowered by adding retarders and rheology-modifying admixtures. Sun et al. [84] examined how a viscosity-modifying admixture of 8% affected the mechanical characteristics of a one-part geopolymer and found that the compressive strength decreased with increasing additive dose. The aeration effect and the development of a thick polymer layer that keeps silicate powder from coming into contact with the activator were blamed for the strength values dropping when a modifier was present [84]. On the other hand, various studies investigated the impact of employing waste material to replace aggregates or a portion of a precursor. Abdollahnejad et al. [183] discovered that substituting burned and unfired ceramic for up to 30% of the GGBS lowered strength values in GGBS-based one-part geopolymers. Depending on the type of aggregate used, replacing natural aggregate may boost or decrease strength [184]. Several studies have shown that, when tested longitudinally (X-direction) to the print direction, printed specimens exhibited higher compressive strength values than cast specimens and other directions. This may be explained by the way the materials move; when they move in the direction of printing, greater compaction is possible after particle placement than when they move in the opposite direction [79]. The printing process characteristics result in anisotropic behaviour depending on the direction of testing [22,88]. The heterogeneity caused by layer interaction may explain the anisotropic nature of printed structures. The 3D-printed product has a denser microstructure than cast concrete due to the tremendous pressure used during extrusion. Despite this, the printed item exhibits greater porosity and weaker connections at the layer interface [185]. Other investigations [113,170] found that cast specimens had greater compressive strength values than 3D-printed one-part geopolymers; this might be explained by the printed samples having higher porosity than the cast samples.

### 6.3. Economic and Environmental Considerations

A primary advantage of employing 3D-printing technology is the reduction of total costs. The use of 3D-printing allows more cost-effective alternatives in terms of material savings, necessary work, and energy. Concrete 3D-printing (3DCP) eliminates the need for formwork, which accounts for 10% of the total cost. As shown in [186], eliminating formwork eliminates the requirement for formwork labour, lowering overall costs by 50 per cent or more. Batikha et al. [187] evaluated several building approaches and discovered that 3DCP is more cost-effective than other construction methods. According to the report, building costs account for 55% of overall costs, while material costs account for 45%. Other investigations discovered that, when utilising a robotic arm for 3DCP, the construction cost consumes 70 percent of the total cost [188,189].

3D concrete printing is the most sustainable construction process, producing less CO_2_ than other methods of construction [187]. Mohammad et al. [190] compared reinforced concrete (traditional method) to 3D-concrete printing and discovered that 3D-printing concrete created around 22% less carbon dioxide emissions. Because 3D concrete printing technology uses more cementitious binder than conventional concrete [10], researchers have focused on finding more environmentally friendly materials instead of cement due to the consumption of approximately 4% of greenhouse gas (GHG) and the release of roughly 8% of total global CO_2_ emissions associated with cement production [191]. According to Yoa et al. [192], using geopolymers in 3D-printing lowers the total carbon footprint associated with producing concrete. Nonetheless, there was a rise in the use of abiotic resources and ozone depletion in the stratosphere. Furthermore, according to Liu et al. [193], compared to the cast OPC sample, cast geopolymers had less of an environmental impact. However, because of the mixture’s increased activator level, it did not print better than OPC. On the other hand, when building a wall, the 3D-printing method’s environmental impact remained constant, whereas the casting technique’s impact changed according to the shape’s complexity [193]. The majority of studies, according to the literature, combined sodium hydroxide (NaOH) and sodium silicate (Na_2_SiO_3_) to create the liquid activator utilised in two-part geopolymers, since NaOH by itself is unable to significantly boost its strength [194]. One-part geopolymers are used instead of two-part geopolymers because there is no need to combine them, which makes the combination more environmentally friendly [195] and [196] provides evidence of the substantial environmental impact associated with the manufacturing of alkali activators. The type of activator determines the amount of CO_2_ emissions produced; Na_2_SiO_3_ has a higher CO_2_ inventory than Ca(OH)_2_. Additionally, depending on the activator used, one-part alkali-activated slag foamed concrete produced 85–93% less CO_2_ emissions than OPC [197]. Making 3D-printable one-part geopolymer mixes using a solid activator can reduce carbon emissions and embodied energy by up to 70% when compared to 3D-printed OPC with a comparable compressive strength value [198].

## 7. Applications

The majority of 3D-printed geopolymers are designed to be used in buildings. In a lot less time than with traditional building technologies, geopolymer technology and additive manufacturing can be used to create structures in a variety of configurations [13,21,79,89,112,113,129]. Because 3D-printed geopolymers cure more quickly and have a higher early strength than OPC-based concrete, they have a considerable advantage as building materials in addition to their substantial environmental and economic benefits. Because of these characteristics and GPs’ strong heat and chemical resistance, 3D-printing of geopolymers is a viable technology for constructing buildings in difficult climatic circumstances, such as lunar construction [122,125]. Attempts to reinforce 3D-printed geopolymer concrete, specifically with micro-cables (Figure 9A) [92,142], improves structural applications. Such composites have reasonably good strength, ductility, and toughness; nevertheless, more research is needed to overcome the limits caused by the presence of weak geopolymer planes between filaments and to improve cable–geopolymer adhesion qualities. Extrusion-based 3D-printing can create highly electrically conductive nanocomposites by adding graphene oxide (GO) nanosheets combined to form a geopolymer. Because of its high chemical activity and polar oxygen functional groups, GO is hydrophilic, which underpins the favourable effects discovered by Zhong et al. in GO/GP nanocomposites [124] and Zhou et al. [199].

The electrical conductivity of the 3D-printed GO/GPs structure shown (Figure 9B) was around 102 S/m due to the decrease of graphene oxide caused by heat treatment of samples at 1000 C. A ceramic matrix composite’s achieved value is relatively high, making it potentially useful as a joule heating thermal source. The addition of GO nanosheets significantly improved the composition’s rheological characteristics, allowing for extrusion-based 3D-printing. The mechanical properties of the nanocomposites improved to a certain critical GO level due to the reinforcing action, beyond which a drop in compressive strength was seen due to nanoparticle aggregation.

Mechanical testing on samples before and after thermal annealing, on the other hand, yielded inconsistent results, which the authors described was a result of pore development from gas evolution when the GO is converted to graphene, as well as of weakening of the graphene layer (see Figure 10) [124]. Later, Zhou et al. [199] discovered that these issues can be mitigated in part by optimising the physical size of GO (50–325 meshes). 3D-printing techniques can be used to create a variety of geopolymeric foams for thermal insulation purposes (see Figure 11).

Alghamdi and Neithalath [94] used a physical foaming method with surfactants to inject air bubbles into the GP matrix to construct such structures. The geopolymeric foams’ high porosities (55–75 percent) and low thermal conductivities (0.15–0.25 W/m-K) make them potentially appealing for use as insulation materials in buildings. Alghamdi and Neithalath [94] used a physical foaming method with surfactants to inject air bubbles into the GP matrix to construct such structures. The geopolymeric foams’ high porosity (55–75 percent) and low thermal conductivities (0.15–0.25 W/m-K) make them potentially appealing for building applications as insulation materials in sandwich wall panels to replace the extensively utilised insulated concrete wall panels. Vlachakis et al. [150] illustrated 3D-printed multifunctional GP sensor-repair patches for concrete structures (Figure 12). 

Utilising the high electrical conductivity of geopolymers supplied by the alkaline solution in their pores and GPs’ capacity to form robust chemical bonds with the rich calcium-based surfaces of OPC concrete, the authors developed a printed GP temperature sensor with a resolution of 0.1 °C and long-term repeatability of 0.3 °C. This example shows how 3D-printing processes in civil engineering can be used for self-sensing and self-healing repairs of critical engineering structures in the field, even though the adhesive bond strength of the printed patch to the concrete substrate was less than that of a typical GP repair material due to the high drying shrinkage [87]. 3D geopolymer printing attempts have been documented. The authors achieved a printed GP temperature sensor with a resolution of 0.1 °C and long-term repeatability of 0.3 °C by utilising the GPs’ capacity to form strong chemical bonds with the rich calcium-based surfaces of ordinary Portland cement (OPC) concrete, in addition to the high electrical conductivity of geopolymers supplied by the alkaline solution in their pores. Although the printed patch’s adhesive bond strength to the concrete substrate was not as strong as that of a standard GP repair material due to the high drying shrinkage, this example highlights the potential of 3D-printing in civil engineering for self-sensing and self-healing repairs of crucial engineering structures in the field. However, surface colonisation by biological micro- and macro-organisms is a feature of 3D-printing GPs and cement-based formulations. Fu et al. [200] demonstrated monoclinic-celsian ceramics generated by high-temperature treatment of (Ba, Sr)-exchanged 3D-printing geopolymer precursors. Controlled geometry and good refractory and dielectric properties pique the interest of those looking for a high-temperature electrical insulator. The mesoporous structure of GPs and the ability to use additive manufacturing to customise the size, distribution, form, and interconnectivity of geopolymer macropores make them appealing for sorption applications. Franchin et al. [201] created porous metakaolin-based geopolymer sorbents for ammonium ion (NH_4_^+^) removal from wastewater using direct ink writing (DIW), also known as robocasting. The sorbents’ hierarchical bimodal (macro and meso) pore structure resulted in positive benefits. The mesoporous aluminosilicate network, in turn, offered a significant ion exchange capacity (>2.6 mg g^−^1). After four cycles, the NH_4_^+^ removal efficiency reached 80%. Luukkonen et al. obtained Ag- or Cu-modified geopolymer water purification filters with disinfecting or catalytic capabilities made by DIW [202]. Compared to identical filters manufactured by direct foaming or granulation, 3D-printed filters had a higher compressive strength. The printed geopolymer filters’ limited water permeability, however, was caused by the low specific surface area, pore volume, and total porosity caused by the drooping and merging of the initial filament layers. This suggests that the rheological properties of the GP paste still need to be improved. Dos Santos et al. [95] used DIW and MK-based geopolymers to build lattice-shaped carriers with mesoporous architectures for enzyme immobilisation. The developed 3D-printed GPs had high permeability coefficients and indicated the potential to sustain *Candida rugosa* lipase immobilisation. The hydrolysis of waste cooking oil as feedstock resulted in a hydrolytic activity of 847 U/g and a yield of 75% free fatty acids, making the utilisation of such 3D-printed GPs as support for biocatalysts fairly intriguing.

Additive manufacturing technologies provide prospects for various advanced energy and environmental applications of 3D-printed geopolymers as catalysts, water purification filters, conductive materials, and smart sensors in addition to their use in the construction industry. Beyond the examples given, we anticipate that research will shortly broaden the spectrum of practical applications of AM technologies, incorporating GPs into new fields. 3D-printed geopolymer nanocomposites are interesting. By adding nanoparticles to geopolymer matrices, composites of this kind gain multifunctionality in addition to improved mechanical and durability qualities. This creates new opportunities for the production of sophisticated geopolymer nanocomposites with special electrical, thermal, electromagnetic, photocatalytic, sensing, and self-healing properties when coupled with the advantages of AM technology. A few of these materials have already been successfully constructed with graphene oxide and graphite nanoplates. The effects of adding various types of nanoparticles to 3D-printable graphene oxides, including carbon nanotubes, nanocellulose, TiO_2_, and ZnO nanoparticles will soon be examined. 3D-printing can be used to create structures with a hierarchical micro, meso, and macro pore size distribution because geopolymer materials may contain nanocrystalline zeolites in addition to the amorphous gel phase, depending on the original component composition and synthesis circumstances [203,204]. This technique combines the functional properties of both the amorphous geopolymeric materials used to immobilise zeolites and the zeolites themselves due to their high surface area and adsorption capacitance with the design freedom of additive manufacturing of geometrically complex structures. These hybrid geopolymer–zeolite materials printed on them have potential applications in filtering, adsorption, and catalysis. These processes, as well as thermal insulation, could benefit from the 3D-printing of foamed geopolymers. Regarding structural design, biomimicry concepts can be used to construct new bio-inspired geopolymer structures with novel functionality.

## 8. Summary and Outlook

This paper examines the design, fabrication, characteristics, and applications of geopolymer materials generated using additive manufacturing. The analysis demonstrates that extrusion-based manufacturing, as the primary method of additive geopolymer manufacturing, can already produce highly complex and multifunctional structures, indicating its prospective applications in environmental remediation, building, and other technical industries. 3D-printing is a sustainable approach to geopolymer material manufacturing, promoting waste reduction, energy consumption reduction, and CO_2_ emission reduction. In addition, 3D-printing technology based on geopolymers confronts hurdles that must be overcome to produce high-performance goods. One of the most important aspects of the quality and accuracy of 3D-printed geopolymers is the design of the mixture. The comparatively high expense, notable role in carbon dioxide emissions, and chemical risk associated with traditional activators warrant additional investigation into more affordable, eco-friendly alternatives. These include sodium carbonate, herbaceous and agricultural biomass ashes, and waste-derived activators derived from silica-rich resources. Utilising industrial waste from mining and building as well as alternate aluminosilicate sources are two ways to advance the development of printing geopolymer combination compounds. Determining the continuity of fresh and hardened state properties of printable graphene sheets and their suitability for different additive manufacturing processes is made more difficult by the compound heterogeneity of precursors. The main limitations of 3D-printing technologies for manufacturing GPs are printing time, printing precision, and the high cost of large-scale printers. Due to its low cost, simplicity, and high-speed production, extrusion-based 3D-printing is now the most widely used approach for additive manufacturing of geopolymer materials. The general problem of additive geopolymer production is anisotropy, which manifests in varied mechanical performance depending on load direction due to a layered approach to structure manufacturing. More research is needed into the effect of air curing conditions on the characteristics of the final geopolymer materials. Studies of the cyclic loading behaviour of printed geopolymer structures are also important in assessing the impact of seismic and artificial dynamic forces. Current restrictions connected with the fabrication of additive geopolymer materials are not insurmountable. However, many studies are still needed to fully grasp the potential of 3D-printed geopolymers in perspective advanced applications.

From multiple angles, the productivity analysis of 3D-printed geopolymers provides optimum values of input parameters to obtain better rheological, physical, and mechanical qualities. Researchers must compromise on selecting desired response values to a limited extent due to several competing natured response measures to have adequate physical and mechanical properties. It is difficult to determine the optimum ranges based on the recommended levels of fresh qualities due to the diverse standards followed by the researchers. On the other hand, the recommended values in the articles provide a good estimate for achieving acceptable workability and buildability. After examining numerous studies, it was discovered that different writers advocated different ideal values for the spread diameter, slump, open duration, and setting time. Furthermore, it has been determined that reinforcement fibres and alkali activators considerably impact the rheological and physical properties of geopolymers. However, there needs to be more research on the rheological requirements of geopolymers for 3D-printing. Many studies have investigated the impact of reinforcement fibre concentration on flexural and compressive strength. For the additive manufacture of geopolymers, most of them have examined synthetic fibres such as glass, carbon, wollastonite, PVA, PP, and steel. In addition, substantial research has been conducted on the impact of post-processing and testing techniques (mainly testing orientation) on mechanical properties. When the effects of testing orientation on mechanical properties are investigated, it is shown that higher compressive strength can be produced when specimens are tested in both the lateral and longitudinal directions. When samples are tested perpendicularly, superior compressive strength is achieved in a few circumstances. Superior flexural strength, on the other hand, can only be obtained in a perpendicular direction. According to our extensive analysis, more research needs to be documented on the impacts of activator and printing conditions on mechanical qualities. Furthermore, the effect of fibre size on mechanical properties has yet to be fully considered. In improving the sustainability of construction processes, a lack of focus on the behaviour of natural fibres in alkaline-activated materials and their influence on mechanical characteristics has been observed.

Regarding the construction and building field, 3D concrete printing is a difficult manufacturing method since work material, printing equipment, and print design are all interconnected terms. Printing process optimisation necessitates the simultaneous progress of all interconnected components rather than optimising them independently. Despite its enormous potential, many problems must be overcome to utilise the 3D-printing of geopolymers in the construction business properly.

The following are the major challenges:The main problem is selecting materials and determining their composition because conventional materials cannot be used in 3D-printing without formwork. Fresh material should have sufficient yield stress to be easily extrudable and preserve its shape after printing. As a result, the materials’ thixotropic qualities must be carefully preserved to achieve a fair trade-off between workability and buildability;The printing orientation of complex structures must be carefully chosen to ensure enough mechanical strength of delicate features and joints of assembly components due to the anisotropic behaviour of 3D-printed concretes, which is not a problem with traditional casting.Due to the weaker green strength of fresh material upon extrusion, overhanging features are difficult to produce with the generally used extrusion-based 3D-printer. Binder jetting or D-shaped technology, on the other hand, offers the capacity to produce complicated shapes due to the bed’s supporting unbounded powder layers. However, in terms of productivity, this technology falls short of extrusion-based printing. More study is needed to improve the mechanical performance of BJ 3D-printed components (green compressive and flexural strength);The poor surface quality of 3D-printed structures is primarily due to layer stacking and dimensional problems, which typically increase post-processing. Inadequate control and higher flowability also cause the printing layers to flow outside, resulting in poor surface quality. If the material’s yield stress and plastic viscosity are kept high to prevent the abovementioned difficulties, surface cracks and voids form, weakening the structure. To a limited extent, inserting trowels into the nozzle aperture can minimise surface roughness.

### 8.1. Concluding Remarks

A few closing thoughts are provided in this study after the productivity of 3D-printed geopolymers has been determined and critical analysis completed:Various activators, additives, and superplasticisers are added to geopolymers to make them workable and printable for large-scale printing because they are not workable in their native state. Two opposing reaction characteristics, workability and buildability, need to be balanced in the new qualities for additive manufacturing. The best parameters for obtaining good fresh and hardened features in geopolymer additive manufacturing are presented in this study. However, these figures are only appropriate for the specific conditions provided in each paper;Different studies used various methods to assess the workability and buildability of geopolymers for extrusion-based 3D-printing. Because of the numerous standards used, defining the ideal range based on the recommended values of the sheets is problematic. However, these recommended settings are an excellent starting point for achieving decent workability and buildability;Based on the slump values advocated in various studies, it can be inferred that superior buildability can be attained in extrusion-based printing with slump values ranging from 8.8 to 20 mm. However, the workability investigation shows that geopolymers with a slump flow diameter (SFD) range of 120–206 mm can be easily produced;The essential concerns for extrusion-based 3D-printing of concrete materials are open time and setting time, which may be regulated by modifying their composition. Many researchers indicated that the optimal open and setting times are 20–80 min and 22.9–90 min, respectively. These numbers may vary for large-scale fabrication since printing large components requires more time, and shorter open and setting times can impair workability;Because of their low green strength, BJ 3D-printed components require post-processing. Post-processing typically entails curing at high temperatures (relative to ambient temperature) or curing in various alkaline solutions for several days, raising the final cost and affecting the sustainability of the 3D-printed geopolymers.

### 8.2. Future Perspective

Following our extensive explanation of the 3D-printing of geopolymer concrete, the following essential challenges must be addressed in future studies:When an alkaline-activated reactive paste is cured, low molecular weight monomers or oligomers are changed from a liquid to a rubbery and solid state. This is because a polymeric network is formed through the chemical reaction between the reactive silico and alluminate groups, which has a significant impact on the three-dimensional structure [20];In order to achieve the required parameters for 3D-printing technology, geopolymer materials need to have a variety of properties, including mechanical, rheological, and physical properties. Key parameters include activator compositions and raw materials made of aluminosilicate, the potential for reinforcing fillers, geometric printing variables, temperature and duration of curing, and post-processing of the 3D-printed specimens [20];Based on a thorough analysis of the literature, it has been determined that reinforcement substantially affects the fresh and hardened properties of 3D-printed geopolymers. The size of the reinforcement fibres must also be evaluated to improve the reinforcement effects and identify a viable steel reinforcement alternative. The effects of natural fibres in alkali-activated geopolymers must also be investigated for additive manufacturing processes;Although printing speed impacts the mechanical characteristics of geopolymers, there has been little research on the implications of printing parameters (printing speed, extrusion rate, etc.) on the ultimate workability and buildability of geopolymers. Aside from material composition, printing parameters can potentially improve geopolymer printing qualities, which can be examined in future studies;Na_2_SiO_3_ and NaOH are required to activate geopolymers, making them less sustainable in terms of cost and environmental impact. Due to the increased cost of alkali activators, this also limits the industrial applications of geopolymers. An inexpensive and environmentally friendly substitute for the alkali activator is desirable to improve the practical applications of geopolymers;Previous research has identified geopolymers and 3D-printing as ecologically friendly materials and a sustainable construction technology, respectively. However, a combined sustainability study of geopolymer 3D-printing is essential in providing a greener direction to the building business;In addition to the one-part geopolymer mixtures that have been successfully generated for 3D-printing, further studies ought to tackle additional challenges. Even though one-part geopolymers have better mechanical properties, further investigation is needed to lessen mechanical anisotropies in 3D-printed filaments by examining the impact of different printing setups and the addition of fibre reinforcement. Furthermore, since most research focuses on mechanical properties, notably compressive strength, it is imperative to investigate the durability of 3D-printed one-part geopolymers. It is also important to investigate how different nanoparticles affect the properties of fresh and hardened materials. The majority of studies focused on how different mix patterns affected the mechanical and rheological characteristics of one-part 3D-printed geopolymers. This is a competitive 3D-printing solution for a range of industrial applications, such as prefabrication and onsite building, thanks to its strong performance and OPC-like preparatory procedures. Because of the quick development of yield stress, open time has seldom been examined and often has a limited window when a solid activator is present. This restricts the use of one-part geopolymers in 3D-printing applications. Therefore, in order to overcome this difficulty and for them to be used in building applications, the effect of different parameters on the open time need to be investigated in order to design a one-part geopolymer combination with a suitable printing window. Additionally, the issue of limited printing open time needs to be resolved by investigating the effects of various retarder and superplasticiser types and dosages, changing the precursor materials, and adjusting the preparation parameters;The viscoelastic behaviour typical of good processability, the mechanical and energetic requirements for material extrusion and deposition, and their final mechanical characteristics can all be understood through the application of a chemorheological approach to the assessment of more ideal processing conditions of a particular alkaline-activated aluminosilicate formulation [20].

Finally, the nonregular aluminosilicate backbone generated by the random copolymerisation of linear and branched sialates favours the formation of an amorphous crosslinked rubber network. The point where this event occurs marks the end of a liquid-like behaviour and announces the transition into a not more processable solid elastic rubber. The still containing reactive oligomers rubber network continues to react, increasing the crosslinking density and allowing to geopolymer vitrification [80,113].

In the reaction mechanism of the alkali-activated materials, the above-mentioned steps could occur in parallel, and their relative weight on the final structure is statistically dependent on the temperature of polymerisation [129].

According to the previously described complexity of the concurrent chemical and rheological phenomena, mathematical modelling and complete chemorheological characterisations become necessary if we want to optimise the conditions of a 3D-printing process where extrusion and deposition are ended before gelation occurs, while the material should be able to self-sustain once deposited [3].

Finally, geopolymer-based additive manufacturing is opening up to new high added-value applications in biomedical fields [20].

## Figures and Tables

**Figure 1 polymers-15-04688-f001:**
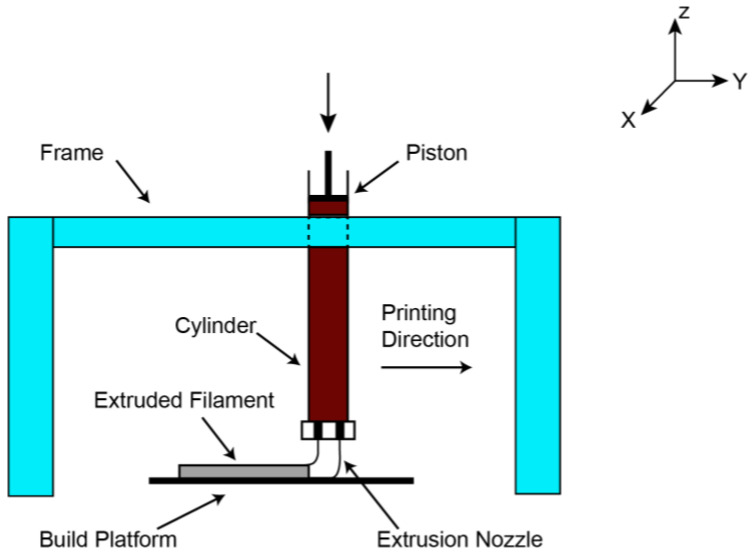
Schematic representation of the extrusion-based 3D-printing process.

**Figure 2 polymers-15-04688-f002:**
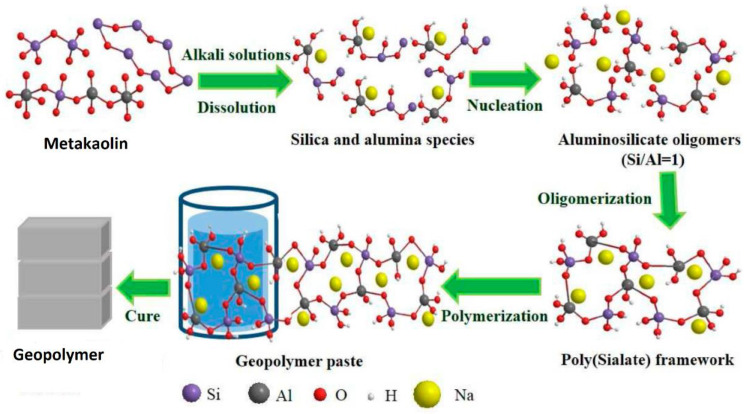
Reaction mechanism of the metakaolin-based geopolymer system [45].

**Figure 3 polymers-15-04688-f003:**
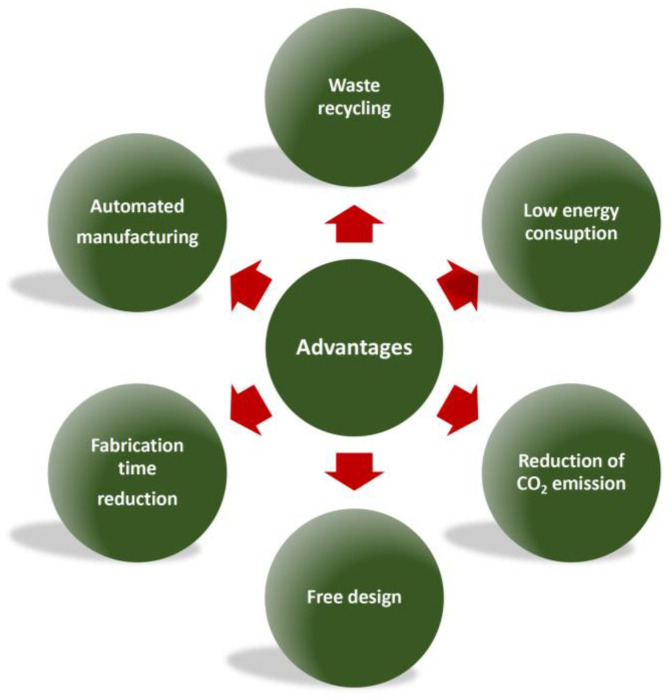
Benefits of geopolymer materials used in 3D-printing technology.

**Figure 4 polymers-15-04688-f004:**
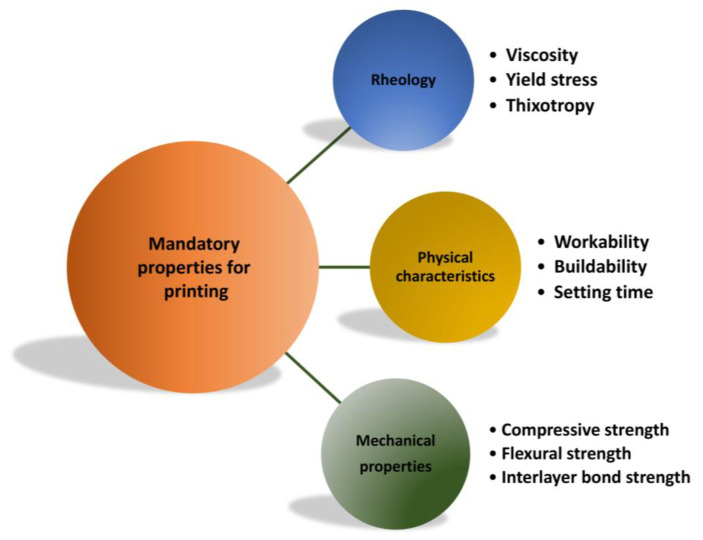
Key properties for 3D-printed geopolymer products.

**Figure 5 polymers-15-04688-f005:**
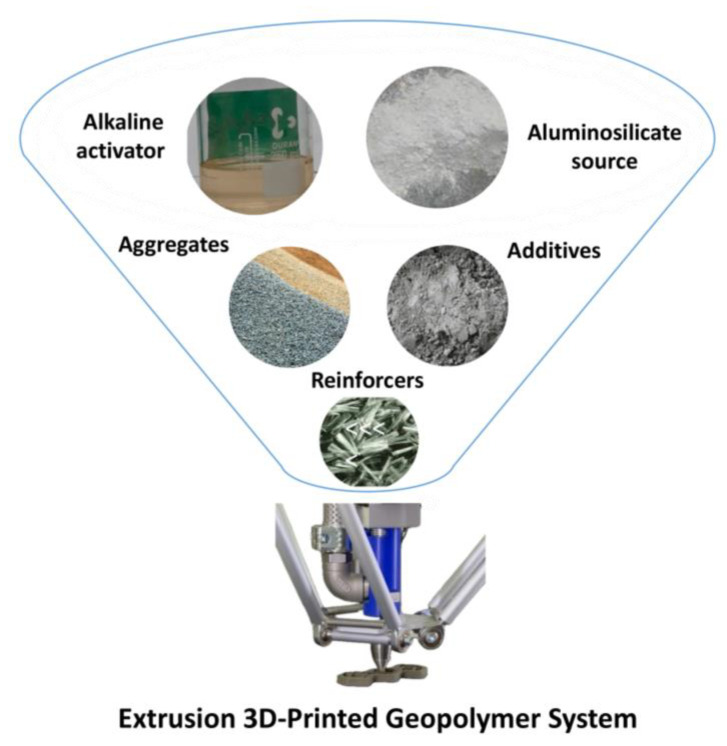
The main additives used in 3D-printing geopolymer technology.

**Figure 6 polymers-15-04688-f006:**
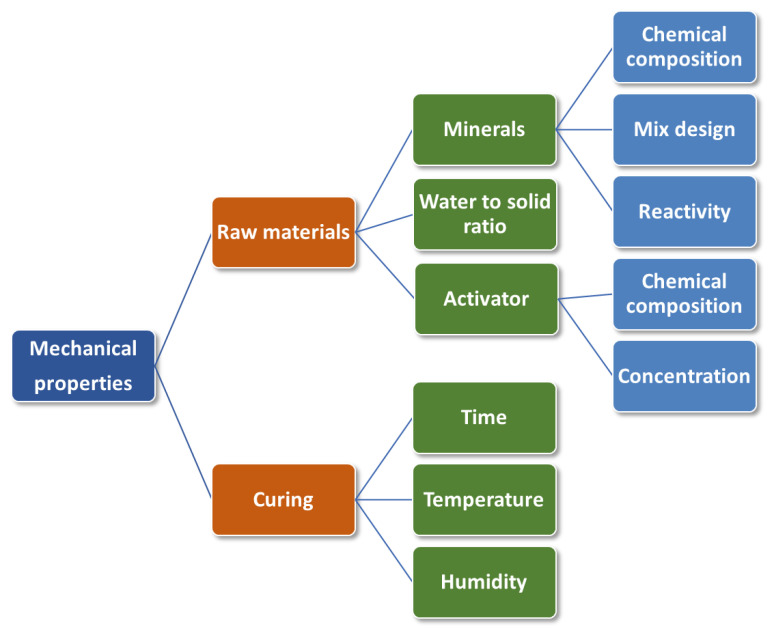
Parameters affecting the mechanical properties of 3D-printed geopolymers.

**Figure 7 polymers-15-04688-f007:**
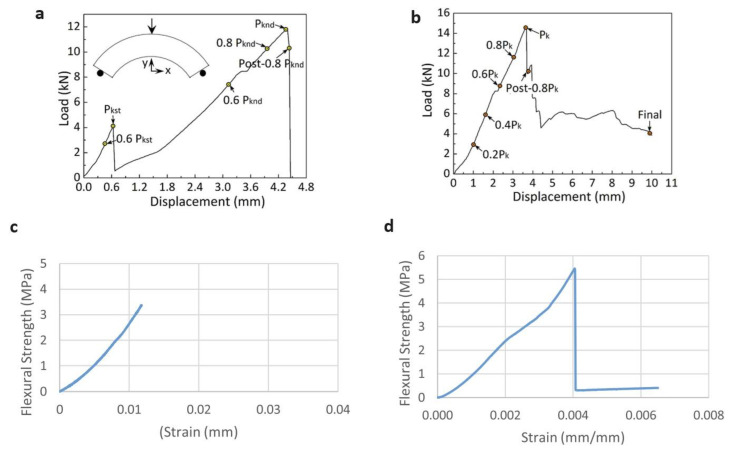
Load–displacement curves of (**a**) nonreinforced and (**b**) continuous micro-cable reinforced arched beams [142]; (**c**) nonreinforced and (**d**) short fibres reinforced systems [129].

**Figure 8 polymers-15-04688-f008:**
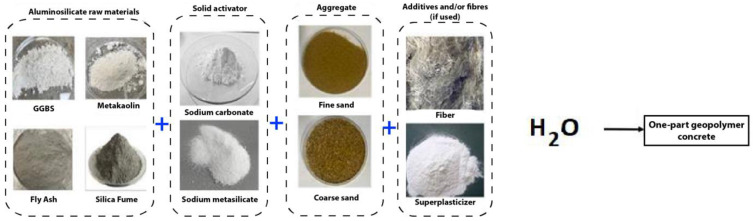
Mix design of one-part geopolymer preparation for 3D-printing application [12].

**Figure 9 polymers-15-04688-f009:**
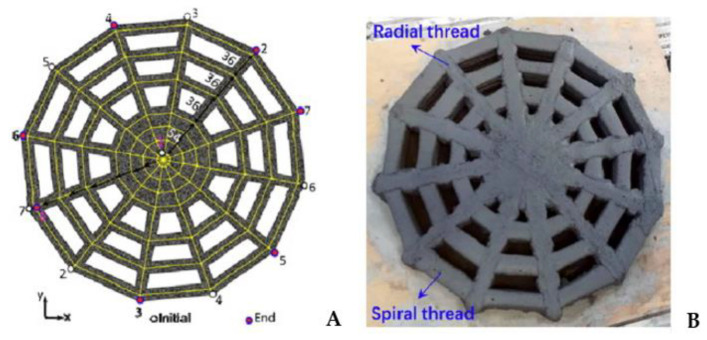
(**A**) Spiderweb-like structure; (**B**) 3D-printed real production of spiderweb-like structure.

**Figure 10 polymers-15-04688-f010:**
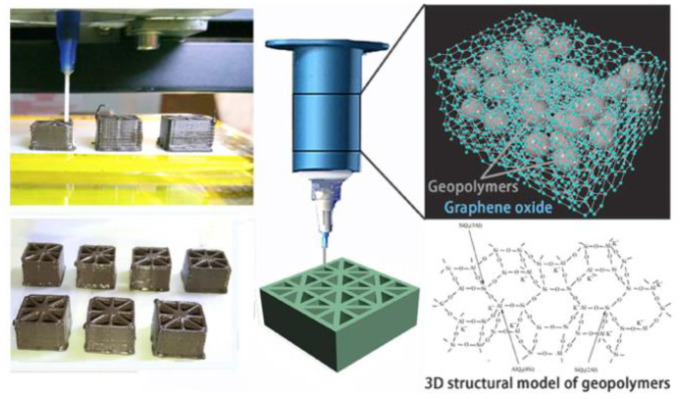
Illustration of the 3D-printing process and some 3D-printed structures. The colours of the printed samples turned from brownish to blackish when the GO loading increased [124].

**Figure 11 polymers-15-04688-f011:**
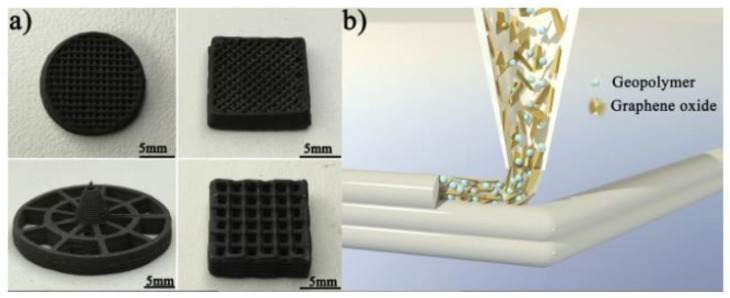
Illustration of 3D-printed structures (**a**), schematic diagram of 3D-printed GOGP (**b**) [199].

**Figure 12 polymers-15-04688-f012:**
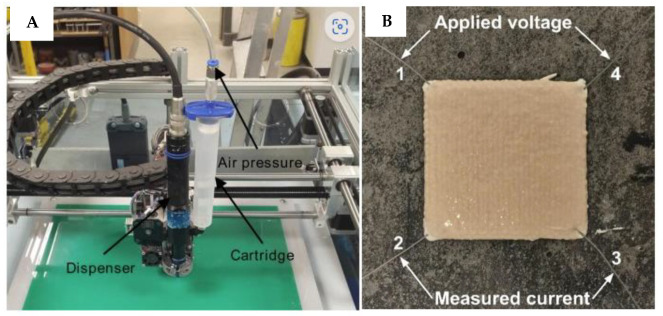
3D-printing setup showing the dispensing unit and an empty pressurised feed cartridge (**A**), the printed sensor on a concrete substrate (**B**) [150].

**Table 1 polymers-15-04688-t001:** Mix design of 3D-printed geopolymer products.

Aluminosilicate Raw Material	Alkali Activator	Aggregates	Rheological and Functional Additives	Reinforcement	Refs.
FA + slag	ASM	Fine silica sand	2-Pyrrolidone	N.F.	[21]
Slag	ASM	Fine silica sand	2-Pyrrolidone	N.F.	[22,25]
FA + BFS + SF	KOH + K_2_SiO_3_	River sand	Actigel and cellulose	N.F.	[13]
FA + BFS + SF	K_2_SiO_3_	Fine river sand	N.R.	N.F.	[53]
FA + BFS	APS + KOH	Fine river sand	N.F.	N.F.	[79]
FA	ASM	N.F.	N.F.	N.F.	[80]
FA + BFS	KOH + K_2_SiO_3_	Fine river sand	Attapulgite clay	N.F.	[81]
FA + BFS	KOH + K_2_SiO_3_	Fine and coarse silica sand	CMC, anhydrous borax	N.F.	[82]
FA + BFS	NaOH + Na_2_SiO_3_	Fine and coarse silica sand	CMC, anhydrous borax	N.F.	[82]
FA + BFS + SF	NaOH + Na_2_SiO_3_	Different grain-sized sands	N.F.	N.F.	[83]
BFS	NaOH + Na_2_SiO_3_	Calcium carbonate	CMC	N.F.	[84]
MK	NaOH + Na_2_SiO_3_	Fine silica sand	N.F.	N.F.	[85]
FA + BFS + SF	ASM	Fine silica sand	Attapulgite clay, SF, slag	N.F.	[86]
MK	NaOH, silica gel	Fine silica sand	N.F.	N.F.	[17]
MK	NaOH + Na_2_SiO_3_	N.F.	Bentonite, microalgal biomass	N.F.	[87]
FA + BFS + SF	K_2_SiO_3_	Fine river sand	Hydroxypropyl methylcellulose	Glass bers (3, 6, 8 mm)	[88]
FA + BFS + SF	NaOH + K_2_SiO_3_	Fine river sand	Attapulgite clay	Glass bers (4 mm)	[89]
FA + BFS + SF	K_2_SiO_3_	Fine river sand	Attapulgite clay	Glass bers (3, 6, 8 mm)	[90]
FA	NaOH, Na_2_SiO_3_	Fine river sand	Carboxymethyl cellulose	PP bers (6 mm)	[91]
FA, BFS, and SF	PSM	Silica sand	N.F.	Nylon, carbon, PE, aramid, steel micro-cables	[92]
FA + BFS + SF	NaOH + Na_2_SiO_3_	Different grain-sized sands	N.F.	Nano-graphite platelets	[93]
FA	NaOH + Na_2_SiO_3_, Na_2_SO_4_	N.F.	SF, OPC, limestone, Lightcrete 02	N.F.	[94]
MK	NaOH + Na_2_SiO_3_	N.F.	Polyethylene glycol	N.F.	[95]
CH, AZS	NaOH + Na_2_SiO_3_	N.F.	Polyethylene glycol PMMA sphere	N.F.	[96]

Abbreviations: N.R. = not reported; N.F. = not found; ASM = anhydrous sodium metasilicate; BFS = blast-furnace slag; PP = polypropylene; CMC = sodium carboxymethyl cellulose; APS = anhydrous potassium silicate; PSM = penta sodium metasilicate; MK = metakaolin; CH = chamotte; AZS = alumina–zirconia–silica; PMMA = poly(methyl methacrylate); SF = silica fume; FA = fly ash.

## Data Availability

Not applicable.

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
