# Peer review of "Geopolymer Materials for Extrusion-Based 3D-Printing: A Review"

_polymers, 2023, doi:10.3390/polym15244688_

Round 1
Reviewer 1 Report
Comments and Suggestions for Authors
The authors mention a research regarding geopolymers for 3D printing based on extrusion, where the materials, the process, and the optimization of these materials are presented, as well as the mechanical, rheological, durability, and printability, among other properties of these geopolymers when they are generated in one and two parts. In addition, applications are presented in various fields and areas of opportunity in the development of these materials. The following suggestions are recommended
Point 1. Figures 4 and 5 on pages 7 and 8 suggest changing the colors of the elements that compose it so as not to be monochromatic and to be more attractive and easier to identify for the reader.
Point 2. In line 83, it is mentioned that the paste must be a thixotropic fluid, but it is not explained why it must exhibit this behavior.
Point 3. Figure 6 shows the main additives in geopolymers. It is suggested that some examples of additive characteristics of each of the classifications are mentioned.
Point 4. Table 3 is mentioned in line 420, however, only Table 1 is presented.
Point 5. Lines 570 and 571 refer to the use of FA in different proportions in the geopolymer; however, it is not indicated what the criterion is or when it is appropriate to use each of these ranges.
Point 6. Lines 653 to 661 (page 16) indicate the use of fibers as reinforcement of the geopolymer matrix. In this section, it is suggested to add a diagram indicating the influence of continuous fibers and short fibers on the mechanical properties.
Point 7. Lines 803 and 804 mentions that the addition of retarders results in better extrudability, but this phenomenon is not justified.
Point 8. Lines 804 to 805 (page 20) refer to the use of superplasticizers; in this section, it is suggested to expand the information regarding rheology and the effects produced in geopolymers.
Point 9. Line 838, it is suggested to increase the content of rheological information on the use of activators.
Point 10. Line 992 refers to the optimal size of the GO, indicating what this dimension would be.
Point 11. The references used in the manuscript are current and relevant to the research. However, references 72 and 75 are the same, as are references 15 and 114 and, finally 16 and 81.
Author Response
Dear Reviewer,
I would like to thank you for your comments and suggestions, which improved the reader's understanding and enhanced the scientific resonance of the manuscript.
Below you can find the answers to your requests.
Point 1.
Figures 4 and 5 on pages 7 and 8 suggest changing the colors of the elements that compose it so as not to be monochromatic and to be more attractive and easier to identify for the reader.
Authors' answer:
We thank the Reviewer for his/her observation. We changed the colours of the figures to become more attractive and easier to identify for the reader.
Point 2.
In line 83, it is mentioned that the paste must be a thixotropic fluid, but it is not explained why it must exhibit this behavior.
Authors' answer:
We thank the Reviewer for his/her observation. An explanation of the phenomenon of thixotropy has been added, as well as the reasons why thixotropy can be used to describe the buildability and 3D structural performance.
Point 3.
Figure 6 shows the main additives in geopolymers. It is suggested that some examples of additive characteristics of each of the classifications are mentioned.
Authors' answer:
We thank the Reviewer for his/her observation. Some examples of additives have been added where suggested by the Reviewer although, the authors describe in detail the types of additives and their role for printability in the following paragraphs.
Point 4.
Table 3 is mentioned in line 420, however, only Table 1 is presented.
Authors' answer:
We thank the Reviewer for identifying the error. The mistake has been corrected.
Point 5.
Lines 570 and 571 refer to the use of FA in different proportions in the geopolymer; however, it is not indicated what the criterion is or when it is appropriate to use each of these ranges.
Authors' answer:
We thank the Reviewer for his/her advice. More information on possible mix designs of geopolymers with fly ash has been added to the discussion.
Point 6. Lines 653 to 661 (page 16) indicate the use of fibers as reinforcement of the geopolymer matrix. In this section, it is suggested to add a diagram indicating the influence of continuous fibers and short fibers on the mechanical properties.
Authors' answer:
We thank the Reviewer for his/her suggestion. We have added some diagrams of load displacement to explain the influence of continuous micro fibres and short fibres on mechanical properties.
Point 7. Lines 803 and 804 mentions that the addition of retarders results in better extrudability, but this phenomenon is not justified.
Authors' answer:
We thank the Reviewer for his/her suggestion. The authors have added additional information on the use of retardants that may affect the early age strength, workability, and setting time.
Point 8. Lines 804 to 805 (page 20) refer to the use of superplasticizers; in this section, it is suggested to expand the information regarding rheology and the effects produced in geopolymers.
Authors' answer:
We thank the Reviewer for his/her advice. In lines 821-834 it is possible to individuate a deep discussion on the use of superplasticizers and rheology:
“Cheng et al. [174] studied the influence of several types of superplasticisers (specifically, polycarboxylate (PC), melamine (M), and naphthalene (N)) on the flowability of calcium carbide residue-waste red brick powder-based one-part alkali-activated materials. The researchers discovered that adding 1.5 percent of PC to the various types considerably boosted the flowability of the mixtures to a level comparable to the OPC mixture. Similarly, Alrefaei et al. [175] investigated the influence of PC, M, and N superplasticizers on the mini-slump performance of FA-GGBS-based one-part geopolymer. They discovered that the flowability of the mixtures increased, with polycarboxylate showing the greatest improvement.
As far as the rheological properties, Panda et al. [80] discovered that increasing the GGBS concentration in an FA-based one-part geopolymer enhanced yield stress, plastic viscosity, and viscosity recovery, owing to the calcium-rich chemical composition of GGBS. Because of the angular morphology of GGBS particles, the packing density increases, improving the thixotropic quality of the mix.”
Point 9. Line 838, it is suggested to increase the content of rheological information on the use of activators.
Authors' answer:
We thank the Reviewer for his/her suggestion. Lines 855-861 contain a detailed explanation of the use of activators and rheology.:
“Aside from the effect of different precursor materials, activator content influences the rheological behaviour of the mix. Muthukrishnan et al. [124] discovered that increasing the activator level increased the mixture's yield stress, viscosity, and thixotropy [124]. Higher activator percentages, on the other hand, may reduce the plastic viscosity of the solution [80]. The composition of the activator (i.e., alkali modulus) also has a significant impact on the rheological properties of the geopolymer. According to [179-180], raising the SiO2/Na2O ratio increased the geopolymer mixture's viscosity and yield stress.”
Point 10. Line 992 refers to the optimal size of the GO, indicating what this dimension would be.
Authors' answer:
We thank the Reviewer for his/her advice. GO dimensions investigated in the study were added.
Point 11. The references used in the manuscript are current and relevant to the research. However, references 72 and 75 are the same, as are references 15 and 114 and, finally 16 and 81.
Authors' answer:
We thank the Reviewer for identifying the error. All mistakes have been corrected.
Reviewer 2 Report
Comments and Suggestions for Authors
This manuscript reviews the-state-of-the-art of geopolymer materials for extrusion-based 3d-Printing, and it is believed to give fundamental understanding of geopolymers during 3D printing. I recommend this paper be accepted after some revisions.
1. Please use standard 3D printing names according to ISO standard.
2. Some latest studies about the extrusion-based 3d-Printing are suggested to cite:
Virtual and Physical Prototyping, 2023, 18: e2245801.
3. Figure 2 is more like a commercial report rather than a scientific paper.
4. Figure 4, Figure 5, and Figure 6 are poor, please redraw this figure.
5. Perspectives in this field should be discussed, and the challenges are also need to analyze.
Comments on the Quality of English LanguageThis manuscript reviews the-state-of-the-art of geopolymer materials for extrusion-based 3d-Printing, and it is believed to give fundamental understanding of geopolymers during 3D printing. I recommend this paper be accepted after some revisions.
1. Please use standard 3D printing names according to ISO standard.
2. Some latest studies about the extrusion-based 3d-Printing are suggested to cite:
Virtual and Physical Prototyping, 2023, 18: e2245801.
3. Figure 2 is more like a commercial report rather than a scientific paper.
4. Figure 4, Figure 5, and Figure 6 are poor, please redraw this figure.
5. Perspectives in this field should be discussed, and the challenges are also need to analyze.
Author Response
Dear Reviewer,
I would like to thank you for your comments and suggestions, which improved the reader's understanding and enhanced the scientific resonance of the manuscript.
Below you can find the answers to your requests.
- Please use standard 3D printing names according to ISO standard.
Authors' answer:
We thank the Reviewer for his/her advice. Names and definitions according to ISO Standard have been added.
- Some latest studies about the extrusion-based 3d-Printing are suggested to cite:
Virtual and Physical Prototyping, 2023, 18: e2245801.
Authors' answer:
We thank the Reviewer for his/her advice. The articles suggested by the Reviewer have been added to the References.
- Figure 2 is more like a commercial report rather than a scientific paper.
Authors' answer:
We thank the Reviewer for his/her observation. We deleted the figure and described the results obtained from the Scopus database in the text.
- Figure 4, Figure 5, and Figure 6 are poor, please redraw this figure.
Authors' answer:
We thank the Reviewer for his/her advice. We have modified Figure 4, Figure 5, and Figure 6 to become more attractive and easier to identify for the reader.
- Perspectives in this field should be discussed, and the challenges are also need to analyze.
Authors' answer:
We thank the Reviewer for his/her suggestion. The Conclusion’s paragraph was further enriched with a discussion on perspectives and challenges.
Round 2
Reviewer 1 Report
Comments and Suggestions for Authors
The authors heeded all the suggestions, only noting a couple of errors on lines 260 (Figure3) and 556 (.[90,94,84]). Therefore, I recommend its publication in this journal.